# Does a Similar 3D Structure Mean a Similar Folding Pathway? The Presence of a C-Terminal α-Helical Extension in the 3D Structure of MAX60 Drastically Changes the Folding Pathway Described for Other MAX-Effectors from *Magnaporthe oryzae*

**DOI:** 10.3390/molecules28166068

**Published:** 2023-08-15

**Authors:** Mounia Lahfa, Assia Mouhand, Karine de Guillen, Philippe Barthe, Thomas Kroj, André Padilla, Christian Roumestand

**Affiliations:** 1Centre de Biologie Structurale, University of Montpellier, INSERM U1054, CNRS UMR 5048, 34090 Montpellier, France; mounia.lahfa@cbs.cnrs.fr (M.L.); assia.mouhand@cbs.cnrs.fr (A.M.); karine.deguillen@cbs.cnrs.fr (K.d.G.); philippe.barthe@cbs.cnrs.fr (P.B.); andre.padilla@cbs.cnrs.fr (A.P.); 2PHIM Plant Health Institute, University of Montpellier, INRAE, CIRAD, Institut Agro, IRD, 34060 Montpellier, France

**Keywords:** protein folding, high-hydrostatic-pressure NMR, MAX-effectors

## Abstract

Does a similar 3D structure mean a similar folding pathway? This question is particularly meaningful when it concerns proteins sharing a similar 3D structure, but low sequence identity or homology. MAX effectors secreted by the phytopathogenic fungus *Magnaporthe oryzae* present such characteristics. They share a common 3D structure, a ß-sandwich with the same topology for all the family members, but an extremely low sequence identity/homology. In a previous study, we have investigated the folding of two MAX effectors, AVR-Pia and AVR-Pib, using High-Hydrostatic-Pressure NMR and found that they display a similar folding pathway, with a common folding intermediate. In the present work, we used a similar strategy to investigate the folding conformational landscape of another MAX effector, MAX60, and found a very different folding intermediate. Our analysis strongly supports that the presence of a C-terminal α-helical extension in the 3D structure of MAX60 could be responsible for its different folding pathway.

## 1. Introduction

While we know from Anfinsen’s work [1] that the 3D structure of a protein is encoded in its sequence, the paths taken by the polypeptide to fold into a well-defined native structure are far from being well understood. This issue is particularly important in the case of proteins which have a similar 3D fold, but very divergent sequences. MAX effectors from *Magnaporthe oryzae* belong to this category: despite having low sequence identity (generally under 25%), they share a conserved structure and topology consisting of a sandwich made of five to six antiparallel β strands [2,3,4,5,6,7,8,9,10]. These MAX-effectors also contain two highly conserved cysteines forming a disulphide bond between the two sheets [5].

*M. oryzae* is a phytopathogenetic fungus that causes the rice blast, the most destructive disease of rice worldwide [11]. During infection, the fungus secretes numerous proteins that act as virulence factors and manipulate host cellular processes to promote fungal virulence and to impair host defenses. An important class of these fungal virulence effectors are small secreted proteins (SSPs) of less than 300 amino acids expressed specifically during infection and without homology to proteins of known activity [12]. Typical fungal pathogens possess several hundreds and sometimes more than a thousand of these SSP candidate effectors.

Based on protein structure, many fungal effectors can be grouped into families [13]. The first such family to be discovered was the MAX (Magnaporthe Avrs and ToxB like) effectors that are present in many fungal phytopathogens and specifically expanded in *M. oryzae* [5]. MAX effectors are massively and specifically expressed during early infection and seem to play critical roles in *M. oryzae* virulence [5,14,15]. Numerous additional families have been identified by high throughput structure modelling of *M. oryzae* candidate effectors [16]. The picture emerging from these studies is that a majority of effectors of the blast fungus can be grouped in more or less extended families of structurally and presumably also evolutionarily related proteins [15].

Recently, we have reported the comparative study of the folding properties of two MAX effectors, AVR-Pia and AVR-Pib [17]. For this purpose, probabilities of contact between specific residues, measured from residue-specific denaturation curves obtained from High-Hydrostatic Pressure Nuclear Magnetic Resonance (HHP-NMR), were used to constrain *Cyana3* calculations [18]. This approach allowed us to characterize the structure and energetics of the folding landscape of the two proteins and to identify a common major folding intermediate consisting in the early fold of the ß3ß4 hairpin. Due to the extreme sensitivity of NMR parameters to the structural environment, and to the fact that a large set of site-specific probes can be studied simultaneously in a multidimensional NMR spectrum [19], NMR spectroscopy has the unique potential to provide a high-resolution, site-specific, time-resolved description of the protein folding reaction. When combined with high-hydrostatic pressure perturbation it can yield unprecedented details on protein folding pathways [20,21,22,23,24]. Following the Le Châtelier principle [25], pressure unfolds proteins because the molar volume of the unfolded state is smaller than that of the folded state; in other words, the volume change upon unfolding $∆V_{u}^{0}$ is negative [26]. Although chemical or thermal denaturation act globally and depend on exposed surface area in the unfolded state, pressure denaturation depends on the elimination of the solvent-excluded internal voids, essentially due to imperfect protein packing [27,28,29,30]. Thus, because the distribution of solvent-excluded voids depends on the protein structure, the pressure-induced unfolding originates from unique properties of the folded state.

In the present study, we applied this strategy to study the folding pathway of MAX60, a new member of the MAX effector family identified through the analysis of the genome of *M. oryzae* with bio-informatic tools [14]. MAX60 shares the topology and the ß-sandwich 3D structure common to all MAX effectors, including the presence of the highly conserved disulfide bridge [31], but displays extremely low sequence identity/homology with AVR-Pia or AVR-Pib. Interestingly, when compared to the structure of AVR-Pia or AVR-Pib, MAX60 differs by the presence of a long C-terminal α-helical extension, protruding from the ß-sandwich. Using pressure denaturation, monitored by 2D NMR, we were able to identify a folding intermediate although urea chemical denaturation suggested a highly cooperative unfolding over the whole protein structure. This folding intermediate results from the early association of the α-helix with the ß1ß6 ß-sheet, also found to be the most pressure-stable secondary structure elements. Thus, the folding pathway of MAX60 was found to be markedly different of that shared by AVR-Pia and AVR-Pib. Our study shows that this result is probably due to the early establishment of tight interactions between the C-terminal α-helix and the ß-sheet. Thus, although marginally affected by important sequence differences, the folding pathway of a protein is likely to undergo drastic changes due to the presence of additional “decorations” outside the core of the molecule, especially when these decorations make close contact with the protein scaffold.

## 2. Results

### 2.1. NMR Resonance Assignment, Solution Structure, and Intrinsic Dynamics

The NMR resonance assignment (including full assignment of the [^1^H,^15^N] HSQC spectra at 32 °C) as well as the solution structure of MAX60 have been published recently [31,32] (BMRB: 34730; PDB: 7ZK0). The structure of MAX60 shares the global fold and a similar topology common to the MAX effectors [5] consisting of a ß-sandwich made of six antiparallel ß-strands (Figure 1). Also, the disulfide bridge characteristic of most MAX effector structures is present at a similar location in the structure of MAX60. Of interest, a helix turn (residues 77–79) replaces the ß5 strand that is present in the structure of most MAX effectors. In addition, the structure of MAX60 displays a long C-terminal extension that mainly comprises a well-defined α-helical conformation (residues 91–102).

^15^N heteronuclear relaxation constants were measured for all amide groups at three different magnetic fields and the corresponding spectral densities were obtained using Equation (1) (Materials and Methods; Appendix A). Lipari–Szabo (Equation (2); [35]) and Extended Lipari–Szabo (Equation (4); [36]) “model-free” approaches were then used to obtain order parameters describing the amplitude of internal motions at the level of the ^15^N-^1^H vectors (see Materials and Methods). Generalized order parameters (S^2^ in Equation (2) and $S_{f}^{2}S_{s}^{2}$ in Equation (4)) and partial order parameters (Equation (4): $S_{f}^{2}$, for fast internal motions, and $S_{s}^{2}$ for slow internal motions) are displayed in Figure 2.

The regular Lipari–Szabo model was used for most of the residues (residues 27–99), including the major part of the C-terminal end (residues 91–99), yielding S^2^ values close to 0.85, and indicating the overall structural rigidity of MAX60. This result also confirms that the C-terminal helix is well-positioned in relation to the ß-sandwich in the 3D solution structure [31]. A value of 6.03 ns was obtained for the global correlation time (τ_c_, Equation (2)) of the molecule from the fit of the spectral densities obtained for these residues, in good agreement with the expected correlation time of this small protein at 32 °C.

An extended Lipari–Szabo model was needed to fit the N- and C-terminal residues (including the last two residues in the C-terminal turn of the helix), suggesting the existence of more complex motions in these flexible regions. Interestingly, when looking to the partial order parameters, we observed a decrease of $S_{s}^{2}$ while the values of $S_{f}^{2}$ remain almost constant and close to 0.80. This behavior has been observed previously in the C-terminal helix of the protein C12A-P8^MTCP1^ [37,38] and suggests that these N- and C-terminal segments remain relatively rigid (limited motion in the picosecond time scale) and are animated by a global hinge motion (in the sub-nanosecond time-scale) that increases from the hinge (the junction with the ß-sandwich for the N-terminus, or the last helix turn for the C-terminus) to the N- or C-terminal end, respectively.

Owing to the multi-field analysis of the relaxation parameters, exchange contribution to $J\left(0\right)$ can be readily measured (Materials and Methods, Equation (3)). A substantial increase of the Φ value (R_ex_ = Φ$\omega_{N}^{2}$) is observed for residues D56-L57 in the ß4 extended strand. Possibly, this could be explained by the proximity of a proline residue (P58) in this segment, with a contribution of cis/trans isomerization. Weaker but significant contributions are also observed for residues in more flexible segments joining the different secondary elements (N39, G53, C73, T90) and also in the ß2 strand (V49 and T51) and in the N-terminal (K92) and C-terminal (I99) turns of the helix.

Finally, amide proton/deuteron exchange rate constants were measured in order to have access to dynamics on longer time scales, reflecting the “breathing” of the secondary structure elements and their “local stability”. H/D exchange was investigated in our experimental conditions (25 mM sodium acetate buffer, pH 4.5) but at 20 °C instead of 32 °C, in order to slow the exchange rate and to make the measurement of H/D exchange rate constants (k_ex_) attainable by real-time 2D NMR. Thus, [^1^H,^15^N] HSQC spectra were recorded with time on a protein sample freshly dissolved in D_2_O (Appendix A; see Materials and Methods for details). An accurate fit with an exponential decay can be obtained for the cross-peak intensity of 42 residues, mostly located in secondary structure elements, over 81 non-proline residues, and the corresponding protection factors (PF) were then calculated [39] from the values obtained for k_ex_ (Figure 3). Next, the local stability of the different secondary structure elements was estimated by the average value of the protection factors (<PF>) calculated over the amide protons involved in H-bond stabilizing the ß-sheets and the C-terminal helix (Figure 3). From this analysis, we observed a hierarchy in local stability for the different secondary structure elements in the order ß1ß2 > ß1ß6 > ß3ß4 > C-terminal helix.

### 2.2. MAX60 Denaturation Studies

#### 2.2.1. Pressure Denaturation

Two-dimensional [^1^H,^15^N] HSQC spectra of ^15^N uniformly labeled MAX60 were recorded at variable pressures (1 to 2500 bar) and at 32 °C (Figure 4). As usually observed, the native form of the protein was found in slow exchange with the unfolded form [24], on the NMR timescale, meaning that the intensity of each native state peaks decreases as a function of pressure, while the intensity of peaks corresponding to the unfolded state, centered around 8.5 ppm in the proton dimension, increases concomitantly (Appendix A). Thus, the simple two-state transition model can be used to interpret the loss of intensity for each native state cross-peak (Materials and Methods, Equation (5)). A total of 65 residues (over 81 non-proline residues) did not give overlapping cross-peaks in the folded state nor in between the folded and unfolded states. In addition, their amide cross-peak displayed sufficient intensity at atmospheric pressure to be accurately fitted to the two-state model, and can be used as a local probe to describe the folding pathway of MAX60. In the conditions of the study, MAX60 displays a moderate stability with an average value for the apparent free energy of unfolding <${∆G}_{u}^{0}$> of 2331 ± 58 cal/mol. The average value of apparent ${∆V}_{u}^{0}$ is 67 ± 2 mL/mol (absolute values of <${∆V}_{u}^{0}$>), in good agreement to what is usually found for a protein of this size, but significantly higher than those found for AVR-Pia (49 ± 7 mL/mol) or AVR-Pib (44 ± 12 mL/mol) [16], suggesting the presence of additional void cavities in the structure of MAX60.

Residue-specific half-denaturation pressures were calculated from these steady-state apparent thermodynamic parameters (P_1/2_ = $∆G_{u}^{0}$/$∆V_{f}^{0}$) (Figure 5). As previously obtained for protection factors, average values <P_1/2_> were calculated for each secondary structure element by averaging the values of P_1/2_ obtained for the amide groups involved in each ß-sheet and the C-terminal helix (Figure 5). These average values could represent a measure of the local stability of the protein. Interestingly, when considering the <P_1/2_> values, the most stable element of the secondary structure appears to be the C-terminal helix, followed by the ß1ß6 sheet, and the ß1ß2 sheet appears to be the least stable. This result is in clear contradiction with the local stabilities of the secondary structure elements obtained from <PF> values.

In order to characterize the folding pathway of MAX60, fractional contact maps were built from probabilities of contact calculated from fractional probabilities of individual residues extracted from the normalized residue-specific denaturation curves (Appendix A) [29,40]. As already reported, the probability of contact for any pair of residues i and j, P_i,j_, at a given pressure was defined as the geometric mean of the fractional probability of each of the two residues in the folded state at the same pressure P_ij_ = $\sqrt{P_{i}\times P_{j}}$ [40] (Figure 6). In the conditions of the study, partial unfolding starts at approximately 1350 bar and concerns mainly the ß1ß2 sheet. At 1400 bar, unfolding progresses to the ß3ß4 sheet, while the ß1ß6 sheet and the C-terminal α-helix start to unfold at 1450 bar. Total protein denaturation is observed at 1500 bar.

In order to obtain a better description of the MAX60 folding process, we used a strategy recently described to decipher the protein-folding conformational landscape of AVR-Pia and AVR-Pib, two other MAX effectors from *M. oryzae* [17]. In this strategy, instead of building fractional contact maps at arbitrary pressures which only provide snapshots of the protein folding process, the entire pressure range is swept (from 1 to 2500 bar, by 25 bar steps), and the probability of contact p_ij_ is calculated at each pressure step for each possible contact between residues. In the present case, two residues were considered “in contact” when the distance separating their Cα atoms (as measured in the NMR structure) is less than 9 Å (272 contacts at 1 bar). This threshold value was retained since it gives sufficient restraints (272) to re-build conformers whose structures are close to the 3D NMR structure of MAX60 (<r.m.s.d.> < 1.5 Å). This step gave 101 lists of probability of contact p_ij_, corresponding to the 100 pressure steps.

For each list of p_ij_ (for each pressure step), restraint lists were built by using a filter ramp ƒ of 251 increments (0 < ƒ < 1, ƒ is incremented by steps of 0.004). This filter ramp was applied to each list of p_ij_: for each increment of ƒ, contacts were considered when p_ij_ ≥ ƒ, and the corresponding Cα-Cα distances measured in the NMR native structure were entered in the restraint list as upper-bound limits. The lower-bound limits were set to the sum of the Van der Waals radii between the two atoms. Repeating this process at each pressure step resulted in 25,250 potential restraint lists.

After removing lists containing either no restraint (corresponding to random-coil unfolded conformers) or all the restraints (corresponding to the native conformers), the remaining lists were used to compute conformer ensembles by the torsion angle dynamics software *Cyana3* [18]: 100 *Cyana3* conformers were calculated for each list, and the conformer with the best target value was retained. These calculations were repeated four times, with four different random draws between lower- and upper-bound limit restraints, yielding four “best” conformers per restraint list and a final total set of 29,064 conformers. This total set of conformers was sorted according to the fraction of native constraints (Q), and to the r.m.s.d calculated between the different conformers at each Q value. MaxCluster software (See Materials and Methods) was used to pool similar conformers into families giving a schematic view of the conformer populations (Figure 7).

This in-depth analysis confirmed the key conclusions of our previous approach: the C-terminal α-helix and the ß1ß6 sheet are the first elements of secondary structure to be formed during the folding of MAX60. Note that during the earlier folding events, the relative positioning of the α-helix and of the ß1ß6 sheet is not well-defined, suggesting a progressive establishment of the native interactions between these two secondary structure elements.

#### 2.2.2. Chemical Denaturation

Two-dimensional [^1^H,^15^N] HSQC spectra of ^15^N uniformly labeled MAX60 were recorded at 32 °C and at increasing urea concentrations (Materials and Methods). A total of 62 residues did not give overlapping cross-peaks in the folded state nor in between the folded and unfolded states, and can be accurately fitted to the two-state pressure-induced unfolding model described in the Materials and Methods (Equation (6); Appendix A). As observed for pressure denaturation, the intensity of each native state’s peak decreases as a function of urea concentration in the NMR sample, while the intensity of peaks corresponding to the unfolded state increases concomitantly, supporting a slow equilibrium on the NMR timescale for each residue between the native and unfolded state during the unfolding process. As reported above for pressure denaturation, a two-state transition model has been used to interpret the loss of intensity for each native state cross-peak. The residue-specific values obtained for the apparent free energy ${∆G}_{u}^{0}$ of unfolding and for the apparent m-values are displayed in Figure 8, with average values of 2316 ± 66 cal/mol and 685 ± 14 cal/mol.M, respectively.

As in the case of pressure denaturation, residue-specific half-denaturation concentrations were calculated from these steady-state apparent thermodynamic parameters ([Urea]_1/2_ = $∆G_{u}^{0}$/m) and average values <[Urea]_1/2_> were calculated for each secondary structure element by averaging the values of [Urea]_1/2_ obtained for the amide groups involved in the ß-sheets and the C-terminal helix (Figure 9). As for <P_1/2_> values, these average values could represent a measurement of the local stability of the protein. When considering the <[Urea]_1/2_> values, the most stable element of secondary structure appears again to be the C-terminal helix, followed by the ß3ß4 sheet, the ß1ß6 sheet, and finally the ß1ß2 sheet. However, the differences between the different values are small (maximum difference < 0.15 M), with relatively important error bars (especially for ß1ß2 and the C-terminal helix). It is therefore not possible to draw firm conclusions about the relative stability of the different secondary structure elements.

Fractional contact maps (Figure 10) were built from probabilities of contact calculated from fractional probabilities of individual residues extracted from the normalized residue-specific chemical denaturation curves (Appendix A).

Contrary to what has been observed for pressure denaturation, increasing urea concentration affected all the secondary structure elements almost simultaneously, suggesting a highly cooperative chemical denaturation of the protein and precluding the identification of any folding intermediate in its folding pathway.

## 3. Discussion

In comparison to the 3D structures of the *M. oryzae* MAX effectors AVR-Pia (PDB: 6Q76 [33]) and AVR-Pib (PDB: 5Z1V [6]), whose folding pathways have been analyzed in a previous study [17], the structure of MAX60 presents two interesting differences. First, the ß5 strand, which is usually involved in the ß3ß4ß5 triple-stranded ß-sheet of the ß-sandwich, is replaced by a helix turn. Note that this strand is reduced to a maximum of only three residues in the crystal structure of AVR-Pia, and forms only few contacts with the ß4 strand, with a maximum of two H-bonds linking these two strands (ß4: E58 CO—HN T71: ß5 and, possibly, ß4: E60 NH—OC I69: ß5). Of course, the helix turn in the structure of MAX60 cannot make any H-bond with the ß4 strand, and thus does not contribute significantly to the stability of the core. Second, MAX60 presents a C-terminal α-helical extension. It is notable that this C-terminal helix does not behave independently from the ß-sandwich: the multi-field ^15^N heteronuclear relaxation study shows that the residues forming this helix do not exhibit additional sub-nanosecond motions and have generalized order parameters S^2^ similar to those found for residues involved in the core of the protein. This is due to multiple interactions between aromatic and other hydrophobic residues belonging to this helix (Y91, Y94, L95) and hydrophobic residues located in the ß6 strand (M89), the ß1 strand (Y30), the ß2 strand (V48, H50), and the loop connecting ß2 to ß3 (Y52) (Figure 11). This hydrophobic patch tightly binds the C-terminal α-helix to the ß-sandwich. Moreover, the stability of the 3D structure of MAX60 is strengthened by the highly conserved disulfide bond C32–C73 which connects the two ß-sheets in the ß-sandwich. This disulfide bridge is present in most of the MAX effectors identified to date, including AVR-Pia (C25–C66) but not AVR-Pib.

One striking result coming from this study is that there is a discrepancy between the conclusions reached by following two different approaches to assess the relative stability of different secondary structure elements in MAX60. The first classical approach consists of measuring the protection factors (PF) of the amide groups involved in the stabilization of the different sheets and of the C-terminal helix. The values of <PF> measured for each secondary structure element indicate that the local stability decreases in the following order: helix < ß3ß4 < ß1ß6 < ß1ß2 (Figure 3). On the other hand, when using half-denaturation pressure (P_1/2_) to estimate the local stability of the same structures, the C-terminal helix becomes the most stable secondary structure element, followed by ß1ß6, then ß3ß4, and ß1ß2 (Figure 5). Note that, in the case of AVR-Pia and of AVR-Pib, the ß3ß4 sheet displays the highest stability, when looking at the averaged <P_1/2_> values measured for the different sheets (Appendix A). The results are less clear when using half-denaturation urea concentration ([Urea]_1/2_) (Figure 9). All the secondary structure elements display a similar relative sensitivity to the denaturant concentration, due to the rather global cooperative chemical denaturation of the protein, even though, again, the C-terminal helix appears to be the most stable. P_1/2_ and [Urea]_1/2_ derive from thermodynamic parameters $∆G_{u}^{0}$ and $∆V_{u}^{0}$ or *m*-values obtained from the fit of residue-specific pressure or chemical denaturation curves, respectively. In this sense, they report directly on the local unfolding, and therefore on the local stability of the structure. Amide proton Protecting Factors are derived from amide proton exchange against solvent (heavy water) deuterons and are related to water accessibility of the amide proton and to the open/closed dynamic equilibrium at the level of the H-bond which involves the amide group. In the case of small proteins like MAX effectors, the burying of secondary structures is negligible, meaning that all amide protons present a relatively similar solvent accessibility. Thus, in the case of MAX60, the equilibrium dynamics between the open/closed forms of the H-bond involving the amide proton remains the main contribution to proton/deuteron exchange. Therefore, the PF measurement reflects more the “breathing” of the element of structure than its unfolding. In the case of MAX60, the C-terminal helix probably presents local dynamics compatible with a fast exchange of the amide proton, but does not reflect its unfolding. By contrast, the dynamics in the sheet are less favorable to proton/deuteron exchange, even though they are less stable and unfold at lower pressures than the α-helix.

An even more striking result concerns the protein folding pathway of MAX60 deduced from pressure denaturation. In a previous study [17], HP-NMR allowed us to conclude that the formation of the ß3ß4 sheet was the earliest event in the folding pathway of AVR-Pia and AVR-Pib, followed by the addition of the ß5 strand to form the ß3ß4ß5 sheet, one of the two triple-stranded ß-sheets of the ß-sandwich (Appendix A). The creation of the second triple-stranded ß-sheet (ß2ß1ß6), which contains the N- and C-termini of the molecule, occurred later in the folding process. AVR-Pia and AVR-Pib display very low sequence identity/homology, and, in addition, AVR-Pib does not have the highly conserved disulfide bridge. Despite this, these two effectors were shown to fold through a similar folding pathway. In the case of MAX60, the folding process starts with the folding of the C-terminal α-helix and the ß1ß6 sheet. The sequence identity/homology between MAX60 and AVR-Pia or AVR-Pib is similar to what has been observed between AVR-Pia and AVR-Pib, and MAX60 has the conserved disulfide bridge. However, MAX60 displays a completely different folding pathway. This might be due to structural differences between MAX60 and AVR-Pia (or AVR-Pib). One difference consists of the replacement of the ß5 strand of AVR-Pia by a helix turn in MAX60. Nevertheless, this strand is considerably reduced in AVR-Pia (and in AVR-Pib) and does not form a canonical triple-stranded ß sheet with ß3ß4. With only one well-defined H-bond between ß5 and ß4, it is difficult to conceive that this very short strand can strongly interfere in the stabilization of the triple-stranded ß-sheet and can contribute in a way to the early formation of the ß3ß4 folding intermediate. The second difference is the presence of a C-terminal α-helical extension in MAX60, compared to AVR-Pia and AVR-Pib, whose 3D structures are restricted to the ß-sandwich forming the conserved core shared by the members of the MAX effector family. Interestingly, the C-terminal helix and the ß1ß6 sheet are the most stable structural elements of MAX60 when looking at their averaged values <P_1/2_>, although it is the ß3ß4 sheet that presents the highest stability in AVR-Pia and in AVR-Pib (Appendix A). This stability is likely to contribute to the earlier formation of these folding intermediates, promoting the aggregation of the other strands toward the final native structure.

One way to test the importance of the α-helical C-terminal extension in the early formation of the folding intermediate found in the conformational landscape of MAX60 would be to study the folding pathway of a mutant without this C-terminal helix (∆Ct-MAX60). However, we have faced difficulties attempting to produce this mutant: in addition to a very low expression yield, the uniformly ^15^N-labeled protein we obtained was very unstable in the conditions of the study (32 °C, pH 4.5) and was contaminated by a significant (≈30%) amount of unfolded species, possibly due to intermolecular aggregation. Nevertheless, we succeeded in assigning the amide groups in the 2D [^1^H,^15^N] HSQC spectrum (Appendix A) through the classical sequential assignment strategy, using 3D [^1^H,^15^N] NOESY-HSQC and 3D [^1^H,^15^N] TOCSY-HSQC. However, continuous irreversible denaturation occurring during the recording of the 3D experiments precluded any attempt at analyzing the folding of ∆Ct-MAX60 with HP-NMR. Nonetheless, this result assesses the importance of the C-terminal α-helix for the stability of MAX60. It should not be seen as a simple “decoration” protruding out of the conserved core shared by the members of the MAX effectors family, but as a key element for the stability of the final scaffold. It is possible that the existence of numerous hydrophobic interactions between this helix and part of the ß2ß1ß6 sheet promotes the early stabilization of the folding intermediate (helix/ß1ß6), avoiding intermolecular aggregation.

One usually considers that if proteins with low sequence identity/homology share a similar 3D structure, this is because they have common key residues conserved through convergent evolution responsible for essential contacts during the folding process, yielding a similar final scaffold. Thus, a similar folding seems to entail a similar folding pathway. This was the case for AVR-Pia and AVR-Pib, two members of the MAX effectors family, sharing a similar structure but low sequence homology. HP-NMR was able to describe a similar folding pathway for these two proteins, involving a similar folding intermediate consisting in the ß-hairpin ß3ß4 [16]. This was also the case for two members of the Immunoglobulin-like (Ig-like) family, I27 and DEN4-ED3, whose folding pathway has been explored using HP-NMR [42,43]. As MAX effectors, Ig-like proteins present a ß-sandwich structure, but with a different topology, involving two four-stranded ß-sheets. I27 is one of the modules found in the I-band of the intra-sarcomere multi-modular protein Titin, involved in the passive elasticity of the striated muscle. DEN4-ED3 is one of the three domains of E protein, the envelope protein from Dengue virus. These two proteins have unrelated function and low sequence identity/homology. Nonetheless, they share a common Ig-like fold structure that displays a similar unfolding process that starts by the disruption of the N- and C-terminal strands on one edge of the ß-sandwich, yielding a folding intermediate stable over a substantial pressure range. In the case of DEN4-ED3, the folding analysis concerned the isolated Ig-like domain, ignoring additional potential stabilizing effects from the other domains of the full-length protein. In the case of I27, the folding of a tandem-repeat was also explored [42], yielding a virtually identical folding intermediate than for the single module. But, in this particular case, there are no interactions between the two identical modules in the repeat, as demonstrated by the perfect superimposition of the HSQC spectra of the single module and of the tandem repeat. Thus, no stabilizing or, on the contrary, destabilizing effect was expected in this construct that might influence the folding process in the tandem. This is not the case in MAX60 where the C-terminal extension is tightly bound to the ß-sandwich core of the protein. In the present study, we have shown that such stabilizing interactions can drastically change the folding pathway of the molecule. Such extensions are common in MAX effector families [32]. These could be N- or C-terminal extensions, unstructured or structured, and include the presence of “tandem” structures with the juxtaposition of two ß-sandwiches. The study of the folding of such proteins is of fundamental interest for our comprehension of protein folding. If the results found for MAX60 can be generalized, it would mean that the folding pathway of a domain in a multi-domain protein does not only depend on its topology, but also on the interactions which can exist between the different domains.

Finally, in contrast to pressure denaturation, chemical denaturation of MAX60 cannot highlight the presence of a folding intermediate in the folding pathway. Instead, a relatively high cooperative unfolding over all of the protein structure is observed. This is probably because chemical denaturation is a harsher method than pressure denaturation, leading to a smoothing of the conformational folding landscape of the protein, and thus erasing potential intermediates. It is important to note that the total pressure unfolding process of MAX60 takes place in a very limited range of pressure: starting at approximately 1350 bar and the unfolding is almost completed at 1500 bar. This suggests the folding energy landscape already has a relatively smooth surface for, where the ß1ß6-Helix intermediate populates a shallow energy well.

## 4. Materials and Methods

### 4.1. Sample Preparation

Cloning of the MAX60 gene, overproduction, and purification of the protein were described in detail by Lahfa et al. [32]. In this construct, the N-terminal signal peptide has been removed from the original sequence, yielding a final protein of 89 residues, numbered from G19 to P107 (PDB: 7ZK0). Using this procedure, the ^15^N-labeled MAX60 protein was prepared by using ^15^NH_4_Cl in a minimal M9 medium and purified by employing affinity chromatography (HisTrap HP affinity column, Cytiva, Freiburg im Breisgau, Germany) and size exclusion chromatography (HiLoad 16/600 Superdex column (Cytiva). The protein was then concentrated to 0.5 mM in the NMR buffer (25 mM Na Acetate, pH 4.5) and stored at −20 °C. The purity was over 95%, as judged by SDS-PAGE.

### 4.2. NMR Assignments and Solution Structure

The virtually complete assignment of ^1^H, ^15^N, and ^13^C NMR resonances through standard 3D triple-resonance assignment as was described by Lahfa et al. [32]. Briefly, backbone and Cβ resonance assignments were made using standard 3D triple-resonance HNCA, HNCACB, CBCA(CO)NH, HNCO, and HN(CA)CO experiments [44] and 3D [^1^H,^15^N] NOESY-HSQC (mixing time 150 ms) and TOCSY-HSQC (isotropic mixing: 60 ms) NMR double-resonance experiments performed on a ^15^N,^13^C-labeled MAX60 sample. Experiments were recorded at 32 °C on a Bruker AVANCE III 800 MHz equipped with a 5 mm Z-gradient TCI cryogenic probe head. The solution structure has been recently solved and published by the same group [32], using *Cyana3* [18] to obtain initial models from NMR restraint sets, that were minimized next with CNS 1.2 according the RECOORD procedure [45] and analyzed with PROCHECK [46].

### 4.3. Relaxation Studies

Relaxation rate constants were measured on Bruker AVANCE III spectrometers operating at three different magnetic fields (14.1, 16.45, and 18.8 T, corresponding to ^1^H frequencies of 600, 700, and 800 MHz, respectively) on a 0.5 mM ^15^N-labeled protein sample. The pulse sequences used to determine heteronuclear ^15^N R_1_, R_2_ relaxation rates, and ^15^N{^1^H}NOE values were similar to those described [47,48,49], and experimental parameters and processing were previously reported in detail for other proteins studied in the laboratory [37,50]. To minimize artifacts, pulse field gradients were inserted during the intervals when the spin system is in a longitudinal spin-order state [51]. The ^15^N longitudinal relaxation rates R_1_ were obtained from 9 standard inversion-recovery experiments, with relaxation delays ranging from 18 ms to 1206 ms. R_1_ data sets were recorded in such a way that the signal intensity decays exponentially to zero as a function of the relaxation delay, thus enabling a simple two-parameter fit. The delay between the 180° ^1^H pulses used to suppress the DD-CSA cross-relaxation was 3 ms. To ensure that water magnetization is minimally perturbed by the application of ^1^H pulses during the T_1_ delays, on-resonance 3-9-19 pulse trains of 180° global flip angle [52] were used with the excitation minimum positioned at the carrier frequency. The ^15^N transverse relaxation rates R_2_ experiments were recorded employing a Carr–Purcell–Meiboom–Gill (CPMG) pulse train [53,54] consisting of four 180° ^15^N pulses and a centered ^1^H pulse, each cycle with a 4 ms duration and the spin-echo period being approximately 1 ms. Eight experiments were acquired, with relaxation delays ranging from 16 ms to 128 ms. R_1_ and R_2_ experiments were recorded with a recycle time of 2 s, coupled with appropriate heating compensation schemes. Moreover, they were performed with relaxation delays arbitrarily chosen in the relevant list (and not in an increasing or decreasing order) so as to prevent any bias that could arise from possible degradation of the main magnetic field homogeneity. Heteronuclear ^15^N{^1^H} NOE were determined from the ratio of two experiments acquired in an interleaved manner for each *t*_1_ increment, without and with proton saturation. The latter is achieved with a train of 120° pulses [55] separated by 20 ms and of a total duration of 3 s. For heteronuclear ^15^N{^1^H}NOEs, special care was taken to avoid large errors that can occur when dealing with protons in fast exchange with the solvent. Accordingly, a carefully optimized water flip-back pulse [56] was added before the last proton 90° pulse in the experiment without saturation. A recycle time of 6 s between scans was used for obtaining a complete recovery of water magnetization and for reducing exchange effects. Moreover, the two experiments with and without proton saturation were acquired in an interleaved manner, FID by FID. A relaxation delay of 30 s was used before the FIDs of the experiment without saturation. For all these experiments, water suppression was achieved by using the WATERGATE scheme [52,57].

NMR spectra were processed with the Gifa 4.22 software [58]. Cross-peak intensities were determined from peak heights [59] using the Gifa peak-picking routine. The relaxation rate constants R_1_ and the R_2_ were obtained from nonlinear fits to mono-exponential functions [60]. The uncertainties due to random errors in the measured heights were deduced from 500 Monte Carlo simulations. The root-mean-square values of noise were evaluated in free-peak regions and used to estimate the standard deviation of the peak intensities.

When the relaxation of the ^15^N nucleus is predominantly caused by the dipolar interaction with its attached amide proton and by the anisotropy of its chemical shift, the relaxation data can be interpreted in terms of the motion of the ^15^N-^1^H vector. Given that the three experimentally determined parameters, R_1_, R_2,_ and ^15^N{^1^H}NOE depend on the spectral density function at five different frequencies [61], the calculation of the spectral density values can be approached by the application of the so-called reduced spectral density mapping, in which the relaxation rates are directly translated into spectral density at three different frequencies [47,48,62,63,64,65]:(1)$$\left[\begin{array}{c} J(0)\\ J(\omega_{N})\\ <J\left(\omega_{H}\right)> \end{array}\right]=\left[\begin{array}{ccc} \frac{-3}{4(3d^{2}+c^{2})} & \frac{3}{2(3d^{2}+c^{2})} & \frac{-9}{10(3d^{2}+c^{2})}\\ \frac{1}{\left(3d^{2}+c^{2}\right)} & 0 & \frac{-7}{5(3d^{2}+c^{2})}\\ 0 & 0 & \frac{1}{5d^{2}} \end{array}\right]\;\times \;\left[\begin{array}{c} R_{N}(N_{z})\\ R_{N}(N_{x,y})\\ R_{N}({H_{z}\to N}_{z}) \end{array}\right]$$
in which $d^{2}={\left(\frac{\mu_{0}}{4\pi }\right)}^{2}\frac{h^{2}\gamma_{N}^{2}\gamma_{H}^{2}}{4\pi^{2}r_{NH}^{6}}$ and $c^{2}=\frac{1}{3}{\left(\gamma_{N}B_{0}\right)}^{2}{\left(∆\sigma \right)}^{2}$.

Where µ_0_ is the permeability of vacuum, *h* is Planck’s constant, γ_H_ (2.6752 × 10^8^ rad·s^−1^·T^−1^) and γ_N_ (−2.711 × 10^7^ rad·s^−1^·T^−1^) are the gyromagnetic ratios of the ^1^H and the ^15^N nuclei, respectively, and ω_H_ and ω_N_ are the ^1^H and ^15^N Larmor frequency, respectively, r_NH_ is the internuclear ^15^N-^1^H distance (1.02 Å), B_0_ is the magnetic field strength, and ∆σ is the difference between the parallel and perpendicular components of the axially symmetric ^15^N chemical shift tensor, estimated to be −170 ppm [50,66]. R_N_(N_z_) and R_N_(N_x,y_) stand for the longitudinal (R_1_) and transversal (R_2_) relaxation rates, respectively. The cross-relaxation rate R_N_(H_z_->N_z_) between ^15^N and its attached amide proton is correlated with NOE and is calculated using: NOE = 1 + (γ_H_/γN).R_N_(H_z_->Nz)/R_N_(Nz). The frequency in the average spectral density <$J\left(\omega_{H}\right)$)> can be taken as equal to 0.87ω_H_ [45].

The model-free approach of Lipari and Szabo [35] was then used to further describe the mobility in terms of specific types of motion. This formalism assumes that overall and internal motions contribute independently to the reorientational time correlation function of ^15^N-^1^H vectors and that internal motions occur on a much faster timescale than the global rotation of the molecule. For an isotropic tumbling protein, one obtains:(2)$$J\left(\omega \right)=\frac{2}{5}\left\{S^{2}\frac{\tau_{c}}{1+{\left(\omega \tau_{c}\right)}^{2}}+\left(1-S^{2}\right)\frac{\tau }{1+{\left(\omega \tau \right)}^{2}}\right\}$$
where τ is the harmonics of the overall and the internal (fast) correlation time which pertains to each residue: $\tau^{-1}=\tau_{c}^{-1}+\tau_{f}^{-1}$. Fast internal motions are characterized by the square of a generalized order parameter *S*^2^, which describes the relative amplitude of internal motions and ranges from 0 to 1, and an internal correlation time τ_f_ for the internal motions.

Multifield measurements of relaxation constants allow us to determine exchange contributions Rex to $J\left(0\right)$ by using the $\omega_{N}^{2}$ linear dependency of the quantity (2*R*_2_ − *R*_1_) according to [67,68]:


(3)
$$2R_{N}\left(N_{x,y}\right)-R_{N}\left(N_{z}\right)={4d}^{2}\left[6J\left(\omega_{H}\right)+4J\left(0\right)\right]+\left[\frac{4}{45}(∆\sigma_{N}{)}^{2}J\left(0\right)+2\phi \right]\omega_{N}^{2}$$


Assuming that $J\left(\omega_{H}\right)$ is negligible with respect to $J\left(0\right)$*,*
$J\left(0\right)$ is obtained from the intercept of the linear représentation of $2R_{N}\left(N_{x,y}\right)-R_{N}\left(N_{z}\right)$ as a function of $\omega_{N}^{2}$ while the slope provides Φ (R_ex_ = Φ$\omega_{N}^{2}$).

For some of the residues, the simple form of Equation (2) is insufficient to fit the whole set of experimental data. This occurs when residues exhibit sub-nanosecond internal motions, remaining in a time window close to 1 ns. In this case, the expression for the spectral density function is extended to [36]:(4)$$J\left(\omega \right)=\frac{2}{5}\left\{S_{f}^{2}S_{s}^{2}\frac{\tau_{c}}{1+{\left(\omega \tau_{c}\right)}^{2}}+S_{f}^{2}\left(1-S_{s}^{2}\right)\frac{\tau }{1+{\left(\omega \tau \right)}^{2}}\right\}$$
with $\tau^{-1}=\tau_{c}^{-1}+\tau_{s}^{-1}$, where $S_{f}^{2}$ and $S_{s}^{2}$ are the square of the partial order parameters for fast (picosecond timescale) and slow (τ_s_, nanosecond timescale) internal motions, respectively. The square of the generalized order parameter *S*^2^, defined as $S_{f}^{2}S_{s}^{2}$, is a measure of the total amplitude of the internal motions. Note that *S*^2^ equals $S_{f}^{2}$ in Equation (2). Equation (4) assumes that the contribution of the fastest motion to the spectral density function is negligible.

The values of the motional parameters of the individual residues can be derived from the fit of the 7 spectral densities obtained from the heteronuclear relaxation rate constants and NOEs measured at three different magnetic fields using equations [2] and [4] implemented in the software DYNAMOF [69] (https://bioserv.cbs.cnrs.fr/SITE/nmr_tools.html, accessed on 15 January 2022). An iterative non-linear least-squares algorithm [60] was employed to further minimize the error function. The “right” model was selected from χ^2^ analysis. As already pointed out [70,71], measurements at several magnetic fields were also important to solve ambiguity in the selection of the model.

### 4.4. Proton/Deuteron Exchange Measurements

For H/D exchange experiments, the protein sample was dissolved in D_2_O buffer (uncorrected pD) at a concentration of 0.5 mM after lyophilization. A series of [^1^H,^15^N]-HSQC spectra were recorded at 20 °C on a Bruker AVANCE III 600 MHz spectrometer (standard ^1^H-^15^N double-resonance BBI probe) with a common measuring time of 7.5 min and a time limit of 112 h. Thirty-two 2D spectra were recorded in the first 4 h, then 8 spectra during the following 4 h (one experiment every half-hour), then 8 spectra during 8 h (one experiment every hour), then 16 spectra during 32 h (one spectra every 2 h), and finally 16 spectra during the last 64 h (one spectra every 4 h). Amide proton protection factors [39] were calculated from the observed exchange rates (k_ex_) obtained from the time dependence of the peak intensities using an exponential decay.

### 4.5. Protein Unfolding

For the pressure denaturation study, 2D [^1^H,^15^N] HSQC were recorded on a Bruker AVANCE III 600 MHz spectrometer (standard ^1^H-^15^N double-resonance BBI probe), at 32 °C and 15 different hydrostatic pressures (1,50, 100, 300, 500, 700, 900, 1100, 1300, 1500, 1700, 1900, 2100, 2300, and 2500 bar). A sample with approximately 0.5 mM concentration of ^15^N-labeled protein was used on a 5 mm o.d. ceramic tube (330 µL of sample volume) from Daedalus Innovations (Aston, PA, USA). A concentration of 0.5 M of guanidinium chloride was added to the sample buffer, in order to obtain complete protein denaturation in the pressure range allowed by the experimental set-up. Hydrostatic pressure was applied to the sample directly within the magnet using the Xtreme Syringe Pump also from Daedalus Innovations. Each pressure jump was separated by 2 h relaxation time, to allow the folding/unfolding reaction to reach full equilibrium. This relaxation time was estimated from series of 1D NMR experiments recorded after 200 bar P-Jump, following the increase of the resonance band corresponding to the methyl groups in the unfolded state of the protein [24]. The reversibility of the unfolding was checked by comparing 1D ^1^H spectrum and 2D [^1^H,^15^N] HSQC recorded at the end of the series of experiment, after returning at 1 bar, with similar experiments recorded at 1 bar before pressurization. No difference was observed between these two sets of experiments.

For the urea chemical denaturation study, 2D [^1^H,^15^N] HSQC were recorded on a Bruker AVANCE III 800 MHz spectrometer (cryogenic ^1^H-^15^N-^13^C triple-resonance TCI probe), at 32 °C and 15 different urea concentration (0, 0.5, 1, 1.5, 2, 2.5, 3, 3.5, 4, 4.5, 5, 5.5, 6, 6.5, and 7 M). Samples with different urea concentrations and approximately 0.2 mM protein concentration were prepared approximately 10 h before recording the NMR experiments. Conventional 3 mm NMR glass tubes (200 µL of sample volume) were used for measurement.

The cross-peak intensities for the folded species were measured at each pressure or each urea concentration, then fitted with a two-state model:(5)$$I=\frac{I_{u}+I_{f}e^{-(∆G_{f}^{0}+p∆V_{f}^{0})/RT}}{1+e^{-(∆G_{f}^{0}+p∆V_{f}^{0})/RT}}$$
in the case of pressure denaturation, or:(6)$$I=\frac{I_{u}+I_{f}e^{-(∆G_{f}^{0}+m[Urea])/RT}}{1+e^{-(∆G_{f}^{0}+m[Urea])/RT}}$$
in the case of chemical denaturation. In these equations, I is the cross-peak intensity measured at a given pressure or at a given urea concentration, and I_f_ and I_u_ correspond to the cross-peak intensities in the spectra of the folded protein at 1 bar or 0 M urea (I_f_ = I_max_, protein in the folded state) and at 2500 bar or 6 M urea (I_u_ = I_min_, protein in the unfolded state), respectively. $∆G_{f}^{0}$ stands for the residue specific apparent free energy at atmospheric pressure or at 0 M urea. $∆V_{f}^{0}$ corresponds to the residue-specific apparent volume of folding for pressure denaturation, while m is related to the steepness of the unfolding transition for chemical denaturation. Half-denaturation pressure (P_1/2_) or half-denaturation concentration ([Urea]_1/2_) were calculated as P_1/2_ = $∆G_{f}^{0}$/$∆V_{f}^{0}$ and [Urea]_1/2_ = $∆G_{f}^{0}$/m.

Native contact maps were obtained by using software CMView 1.1 [41] (http://www.bioinformatics.org/cmview/; accessed on 25 March 2020) with a threshold of 9 Å around the Cα of each residue, using the best structure obtained for GIPC1-GH2 among the 20 refined ones.

The conformational landscape of MAX60 was obtained through a strategy described in detail in [16]. Briefly, this strategy integrates information coming from two main lists:-A list of Cα-Cα distance upper bounds with a cutoff of 9 Å generated by CMView from the PDB structure 7ZK0 of MAX60. In addition, lists of backbone dihedral restraints (Φ/Ψ at ± 10°) were also derived from the structures.-A list containing the probability *p*_i_ to find a residue *i* in a folded state, obtained from the normalized experimental residue-specific denaturation curve obtained for residue *i*. These curves are obtained from the fit of the intensity decrease with pressure of HSQC cross-peak of residue *i* with Equation (5).

For a given pressure, 250 lists of contacts were established through filtering each native Cα-Cα contact by increasing cut-off values (ƒ) obtained from a ramp (ƒ = 0.004 to f = ƒ) by step of 0.004: a native contact between residues *i* and *j* is included in the list for a given ƒ value if (according to Equation (4)):


$$\sqrt{p_{i}\cdot p_{j}}\geq f$$


Constraint lists having zero or all the native contacts were discarded. The Cα_i_-Cα_j_ distance measured in the structure of MAX60 was used as an upper bound limit to restrain the distance between residue *i* and *j* in *Cyana3* calculations. In addition, the backbone Φ_i_/Ψ_i_ and Φ_j_/Ψ dihedral angles measured in the solution structure were used as constraints (±10°) only for residues in the ß-strands or the α-helix to further restrain the available conformational space of residues involved in contacts during calculations. This procedure was repeated at each pressure, from 1 to 2500 bar, with 25 bar steps.

MaxCluster (http://www.sbg.bio.ic.ac.uk/~maxcluster/download.html, downloaded 15 May 2019) software was used for the clustering of the 29064 conformers obtained through this procedure. The clustering must perform two goals: (i) global shape recognition and clustering at low Q values and (ii) more local r.m.s.d. clustering at high Q values. Accordingly, we adapted MaxCluster parameters that perform well with these two tasks. Since we performed 4 *Cyana3* calculations for each constraint list, yielding 4 conformers per list, we choose the minimum number of conformers to form a cluster to be 5 (default value).

## 5. Conclusions

The relevance of pressure-induced unfolding mechanisms to the unfolding pathway of a protein under physiological conditions remains an important question. As is often the case for the study of biological mechanisms, the in vitro study of protein folding imposes to drastically simplify the system. The use of dilute samples is appropriate (and mandatory!) for the thermodynamic analysis of the data, but it is very far from the very crowded physiological cell conditions. In addition, within living cells, protein folding might not be an equilibrium process, but might occur in a vectorial manner as the nascent polypeptide emerges from the ribosome, during translation, into the complex cellular environment [72,73,74]. The simple presence of a nearby surface may result in entropic effects on the thermodynamics of the folding process, while interactions with the surface may preferentially stabilize folded or unfolded states, given the high local concentrations resulting from tethering [75]. The co-translational acquisition of folded structure has been demonstrated for a number of systems by a very wide range of biophysical, spectroscopic, and imaging techniques, including NMR spectroscopy [76]. Nevertheless, this co-translational view of the folding mechanism cannot be generalized to all the proteins. In the topology of the MAX effector family, the N- and C-terminal strands form a ß-sheet in the ß-sandwich (ß1ß2ß6 sheet, Figure 1), meaning that they should have been both synthetized before their association. This supports a post-translational mechanism for their folding process. In such cases, the in vitro folding pathway becomes probably more realistic and closer to what happens in vivo, even though effects due to the crowded environment may perturb the kinetics or thermodynamics of the folding reaction, potentially altering the folding pathway from that which would be observed in equilibrium studies of the isolated protein.

Moreover, in some cases, the physiological relevance of the folding intermediates found in the protein folding pathways is a clear demonstration of the biological soundness of in vitro unfolding studies. In a previous study [42], we have shown that High-Pressure NMR and force spectroscopy revealed the same folding intermediate in the folding pathway of the Titin I27 module (whereas they involve different mechanical perturbations!). In this intermediate, the N-terminal strand is detached from the ß-sandwich of the Ig-like domain. It has been proposed that this partial and reversible unfolding of the I27 module might be responsible for the contribution of Titin to the striated muscle elasticity.

## Figures and Tables

**Figure 1 molecules-28-06068-f001:**
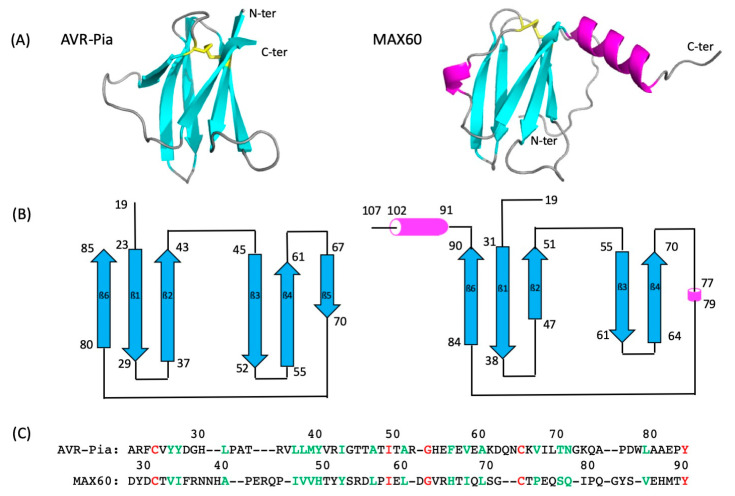
Comparison of the structure of AVR-Pia and MAX60. (**A**) Cartoon representation of AVR-Pia (PDB: 6Q76 [33]) (**left**) and MAX60 (PDB: 7ZK0 [31,32]) (**right**) 3D structures. The ß strands are colored in cyan, the helical segments of MAX60 in pink. The SS-bridge is shown by yellow sticks. (**B**) Topological diagram of the secondary structure of AVR-Pia (**left**) and MAX60 (**right**). The blue arrows and the pink cylinders stand for ß-strands and helices, respectively. (**C**) Sequence alignment (TM-align [34]) of the two proteins (ß-sandwich only). Residues colored in red correspond to identical residues in the two sequences (sequence identity ≈ 6%). Residues colored in green correspond to homologous residues between the two sequences (≈27% sequence homology). The numbers on top of each sequence correspond to the sequence numbering of each protein.

**Figure 2 molecules-28-06068-f002:**
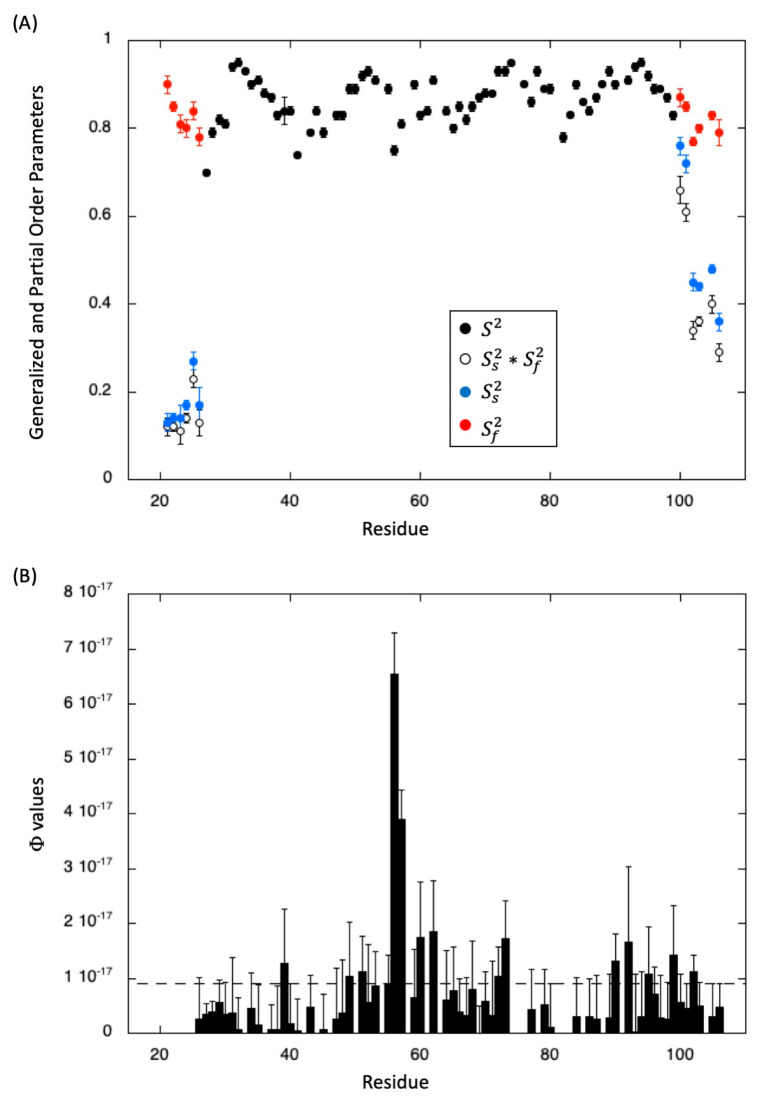
Intrinsic dynamics of MAX60. (**A**) Generalized ($S^{2}\;and\;S_{s}^{2}\times S_{f}^{2}$) and partial ($S_{s}^{2}\;and\;S_{f}^{2}$) order parameters obtained from Lipari–Szabo (Equation (2)) and Extended Lipari–Szabo (Equation (4)) models, respectively (See Materials and Methods). (**B**) Φ values obtained through the multifield measurements of relaxation constants (Equation (3), see Materials and Methods). The dashed line corresponds to the average value.

**Figure 3 molecules-28-06068-f003:**
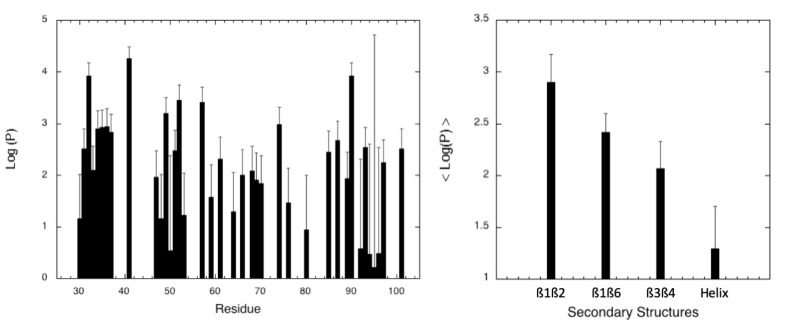
Amide proton/deuteron exchange rates. (**Left**): Protection factors (PF), obtained from amide H/D exchange rates, versus the protein sequence. (**Right**): average values of the protection factor (<PF>) calculated over the amide protons involved in H-bond stabilizing each secondary structure element in MAX60.

**Figure 4 molecules-28-06068-f004:**
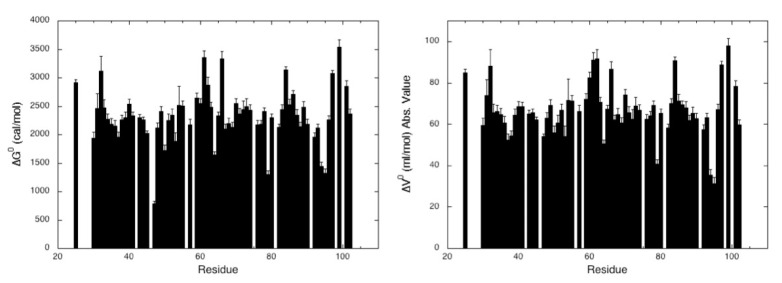
Steady-state thermodynamic parameters measured for MAX60. Residue-specific values for the apparent free energy of unfolding ${∆G}_{u}^{0}$ (**left**) and the apparent volume change of unfolding (absolute values) ${∆V}_{u}^{0}$ (**right**) plotted versus the protein sequence. These apparent thermodynamic parameters were obtained from the fit of the pressure-dependent sigmoidal decrease of the residue cross-peak intensities in the HSQC spectra with Equation (5) (Materials and Methods).

**Figure 5 molecules-28-06068-f005:**
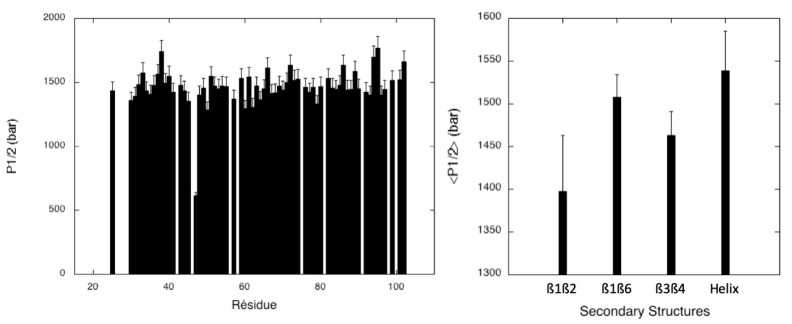
Half-denaturation pressure (P_1/2_) values measured for MAX60. (**Left**): half-denaturation pressure (P_1/2_), obtained from the ratio $∆G_{u}^{0}$/$∆V_{f}^{0}$ extracted from each residue-specific denaturation curve, versus the protein sequence. (**Right**): average values of the half-denaturation pressure (<P_1/2_>) calculated over the amide groups involved in each secondary structure element in MAX60.

**Figure 6 molecules-28-06068-f006:**
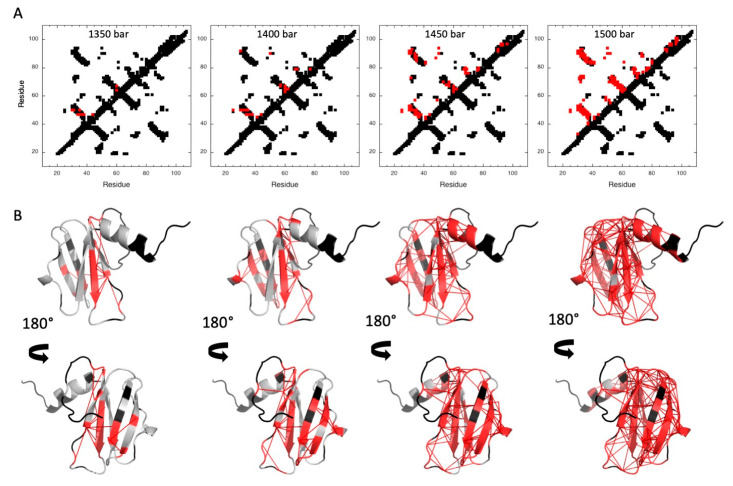
Pressure denaturation of MAX60. (**A**) Fractional contact maps built from the NMR structure of MAX60 at 1350, 1400, 1450, and 1500 bar and 305 K, as indicated. Contacts below the diagonal have been calculated with CMview [41] and correspond to residues where the distance between corresponding Cα atoms is lower than 9 Å. Above the diagonal, only the contacts for which fractional probability can be obtained have been reported. In addition, contacts have been colored in red when contact probabilities P_ij_ lower than 0.5 are observed. (**B**) Visualization of the probabilities of contact on ribbon representations of MAX60 at 1350, 1400, 1450, and 1500 bar. The red lines represent contacts that are significantly weakened (P_ij_ ≤ 0.5) at the corresponding pressure. Residues involved in these contacts are also colored in red.

**Figure 7 molecules-28-06068-f007:**
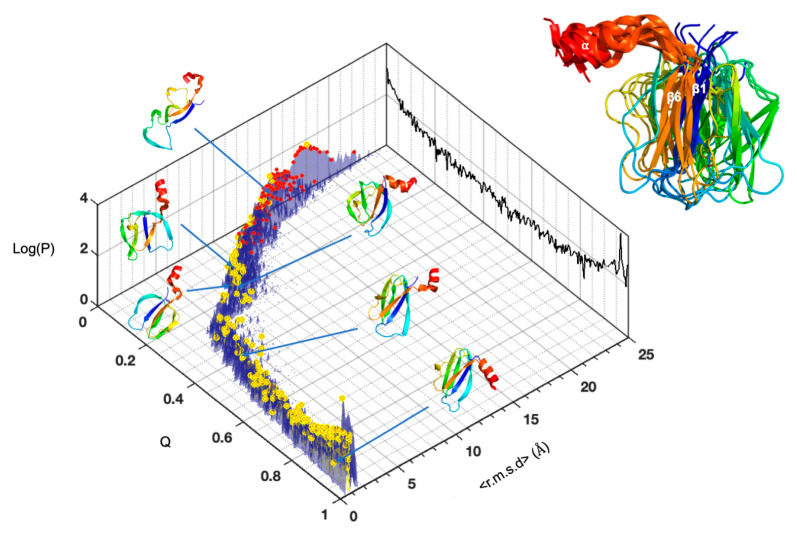
The conformational folding landscape of MAX60. Distribution of the populations of conformers (Log scale) in relation to their <r.m.s.d.> and their fraction of native constraints (Q) for MAX60. Cluster centroids are indicated by the red dots. The centroids for the most populated clusters are indicated by the yellow dots. The total conformer populations (Log scale) versus Q are projected on the back planes (black color). Characteristic conformers are displayed along the conformational space. The early folded intermediate (C-terminal helix + ß1ß6 helix) is shown in the insert as the cartoon superimposition of the conformer centroids in the most populated clusters at Q = 0.2. Conformers centroids are in rainbow colors from N-ter (blue) to C-ter (red) and the ß1and ß6 strands and the C-terminal helix are labelled.

**Figure 8 molecules-28-06068-f008:**
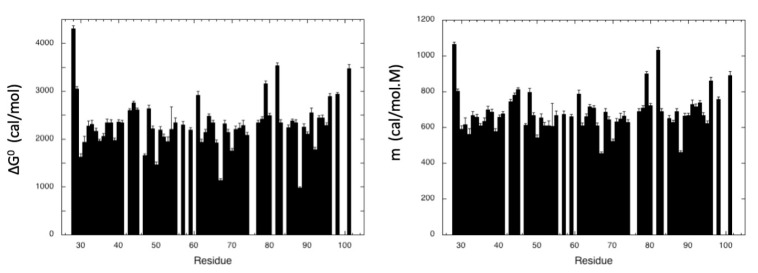
Steady-state thermodynamic parameters measured for MAX60 from residue-specific urea denaturation curves. Left panel: residue-specific absolute values of the apparent free energy of unfolding ${∆G}_{u}^{0}$ plotted versus the protein sequence. Right panel: residue-specific values of the apparent m-values plotted versus the protein sequence.

**Figure 9 molecules-28-06068-f009:**
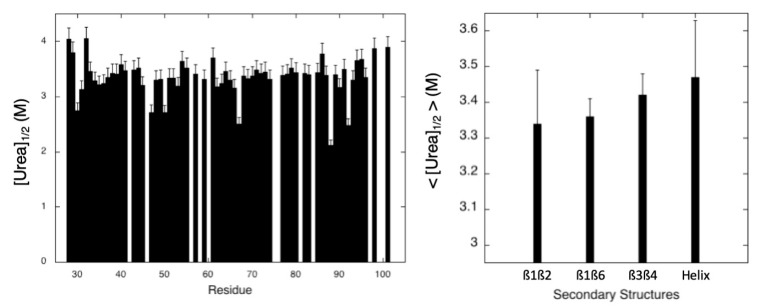
Half-denaturation urea concentration ([Urea]_1/2_) values measured for MAX60. (**Left**): half-denaturation pressure ([Urea]_1/2_) obtained from the ratio $∆G_{u}^{0}/$m extracted from each residue-specific denaturation curve, versus the protein sequence. (**Right**): average values of the half denaturation urea concentration (<[Urea]_1/2_>) calculated over the amide groups involved in each secondary structure element in MAX60.

**Figure 10 molecules-28-06068-f010:**
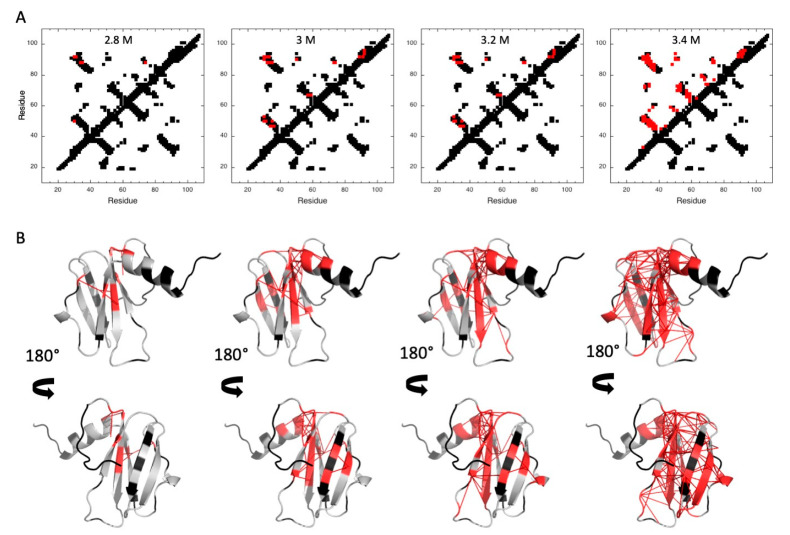
Chemical denaturation of MAX60. (**A**) Fractional contact maps built from the best solution structure obtained for MAX60 and at 2.8, 3, 3.2, and 3.4 M urea, as indicated. Contacts below and above the diagonal are displayed following the same rules as in Figure 6. (**B**) Visualization of the probabilities of contact on ribbon representations of MAX60 and at 2.8, 3, 3.2, and 3.4 M urea. The red sticks represent contacts that are significantly weakened (P_ij_ ≤ 0.5) at the corresponding pressure. Residues involved in these contacts are also colored in red.

**Figure 11 molecules-28-06068-f011:**
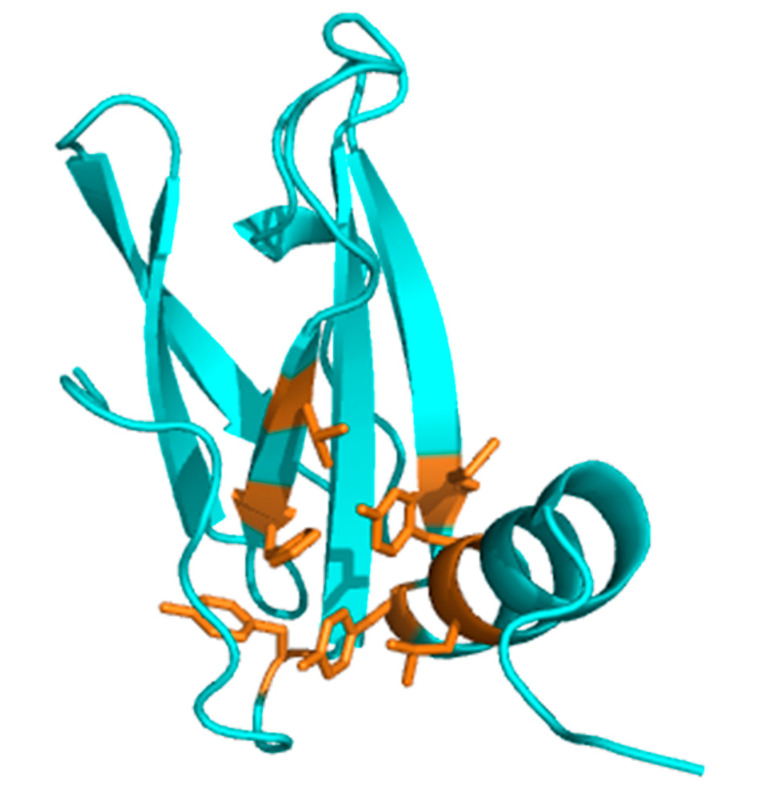
Hydrophobic contacts between the C-terminal α-helix and the ß-sandwich of MAX60. Ribbon representation (cyan) has been used for backbone atoms, sticks for the hydrophobic residues (orange, side chains).

## Data Availability

Not applicable.

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
