# Peer review of "Does a Similar 3D Structure Mean a Similar Folding Pathway? The Presence of a C-Terminal α-Helical Extension in the 3D Structure of MAX60 Drastically Changes the Folding Pathway Described for Other MAX-Effectors from *Magnaporthe oryzae"

_molecules, 2023, doi:10.3390/molecules28166068_

Round 1

Reviewer 1 Report

The manuscript by Roumestand and coworkers describes an interesting study on the folding pathway of a Max effector, namely Max60. The manuscript is generally sound, the used methodologies are perfectly suited and well used. The research topic is absolutely interesting and should be more investigated. The results are well presented and convincing. Nonetheless one of the main results, i.e. the protein flexibility coupled to a high structural stability, in the helical region, should be better commented and referred to previous studies, for example those showing increasing of flexibility  in thermostable protein. Furthermore, the comparison of MAX60 folding mechanism with those of I27 and DEN4-DEN3 should be moved from conclusions to the discussion, reorganizing the final paragraph.

Reviewer 2 Report

Work by Lahfa et al is devoted to the problem of protein folding pathways. While there is much knowledge regarding the properties of protein folded states, little is known about the pathways to how protein finds its final conformation. This results in the absence of techniques that would allow the prediction of the protein refolding probabilities under various conditions, which hinders substantially the development of biotechnology and structural biology. The present study is done using high-pressure NMR spectroscopy which is a state-of-the-art technique. The study is thoroughly done, and the paper is well-written and illustrated. In my opinion, it can be published after the revision.

However, I have several questions/comments, that, to my opinion, need to be answered.

  1. The major question for me is the problem of individual free energies. The pattern described by the authors agrees with the all-in-one transition, but the authors claim that individual residues have their individual free energies of folding and half-pressures.  Shouldn't there be a third (fourth and etc) state then (not just 100% folded and 100% unfolded, but partially folded as well)? Is it possible, that while some part of the protein is already unfolded, the other part of the protein still reveals the signals at the same positions as in the 100% folded or unfolded states? Could it be that the apparent differences in dG are caused by the different transverse/longitudinal relaxation of peaks corresponding to the folded and unfolded states? It is rather obvious that the population of states and not the peak intensities should be standing in equation (5), and these are quite different due to the effects of transverse relaxation on the magnetization during the pulse sequence. The easiest way to check is to answer the question - is the sum of If and Iu retained at all the experimental points. If not, then a correction should be introduced for all the residues so that Iu + a*If = const.  In my opinion, it is essential to do such kind of check, at least for several residues that currently demonstrate the reliable difference in free energies of unfolding.

  2. Stability of the secondary structure elements - is the difference observed between the elements statistically reliable? Especially, this point is important for Figure 9, the difference there seems to be less than the standard deviation.

  3.  Authors always claim that the intermediates that they observe when increasing pressure describe the folding pathway of the protein. Is it really so? Are there studies, which allow concluding that the pressure-induced unfolding and the on-ribosome or in vitro folding of the protein occurs via the same intermediate states? If not, I believe that this problem at least needs to be raised in the discussion.

  4. Finally, it looks like the discussion could benefit from the figure, which would depict the difference between the folding of MAX60 and other effectors, which are discussed in the text.

Some minor comments:

  1. l. 52 "To this aim" could be changed to "for this purpose"

  2. l62. "It can yields" - please change to "it can yield"

  3. l.216 "could represent a measure"

  4. l.379 - "numerous" I  suggest changing to multiple

  5. please specify the used construct in 4.1 - the residue numbers according to Uniprot, corresponding to the final protein sample.

  6. l.510 please change "inductions" to "fields"

  7. l. 519 please indicate the recycling delay

  8. l. 561 - "were corrected from" should be rephrased, and the whole sentence is quite difficult to understand

  9. l. 572. close to 1 ns or to the overall tc value?

  10. l. 711 "common on" please change to "common in"

Reviewer 3 Report

Protein folding/unfolding and folding pathways are evergreen problems, therefore this paper is interesting. Some ways are known to monitor the details of unfolding by shifting the folded/unfolded equilibria via changing the external conditions:  temperature, chemical agents or pressure. Pressure induced unfolding comes of age since high pressure NMR tubes are available and NMR is a relatively sensitive tool for isotope labelled proteins. In the present work both pressure and chemical unfolding were investigated and dynamics was studied by multiple field 15N-relaxation and H/D exchange. Interestingly, pressure induced unfolding was followed in the presence 0.5M guanidium chloride, that means a combined effect of pressure and chemical effect.

Still, there are caveats of each method. Increased pressure increases water viscosity, that may reduce the sensitivity of the the 15N HSQC experiments and thereby reduce the peak integrals at high pressure (T2 relaxation effects, pulse calibration changes caused by added salt). Were these problems observed and considered ? How was the reversibility of unfolding checked ?

Spelling, typing etc. errors should be fixed. Some notes, questions and amendements are mentioned below at the assigned raws:

259:    why 9 Angstrom ?

511:     600, 700 and 800

513:    please provide Bruker pulse sequence names, and important parameters

534-535:  10 8, 10 7 upper case exponents

542:    RN,  N must be in lower case

587:    is the DYNAMOF code publicly available ?

593:    in D2O or H2O ?, cf. fig S2

611-612:  was therefore a mixed chemical (with 0.5M guanidium chloride) / pressure unfolding ?

627: cross-peak intensities (amplitudes) or integrals ?

Fig S3: selected residues should be marked on 2D maps

Fig S5: selected residues should be marked on 2D maps

Fig S6: why are the residue specific errors significantly higher if compared to Fig 5. of MAX60 ?

Fig S8: would be nice to connect the folded/unfolded states of the three selected residues by arrows (I35, Y53 and V64) on the 2D map.

English proofreading is a must. Some errors are mentioned below.

385:    is present

412:    derived

606:   were
